# Antioxidant and Angiotensin-Converting Enzyme (ACE) Inhibitory Activities of Yogurt Supplemented with *Lactiplantibacillus plantarum* NK181 and *Lactobacillus delbrueckii* KU200171 and Sensory Evaluation

**DOI:** 10.3390/foods10102324

**Published:** 2021-09-30

**Authors:** Eun-Deok Kim, Hyun-Sook Lee, Kee-Tae Kim, Hyun-Dong Paik

**Affiliations:** 1Department of Food Science and Biotechnology of Animal Resources, Konkuk University, Seoul 05029, Korea; ked0823@konkuk.ac.kr (E.-D.K.); richard44@hanmail.net (K.-T.K.); 2Department of Foodservice Management and Nutrition, Sangmyung University, Seoul 51767, Korea; leehs9292@smu.ac.kr

**Keywords:** probiotics, yogurt, antioxidant, ACE inhibition, sensory evaluation

## Abstract

This study was carried out to develop a functional yogurt with inhibitory effects on angiotensin-converting enzyme (ACE) and antioxidant activity using various probiotic strains. Yogurts were prepared using a commercial LAB freeze-dried product and probiotics.Yogurt with only commercial LAB product as control group (C) and probiotics supplemented with *Lacticaseibacillus rhamnosus* GG KCTC 12202 BP, as a reference group (T1), *Lactiplantibacillus plantarum* KU15003 (T2), *Lactiplantibacillus plantarum* KU15031 (T3), *Lactiplantibacillus plantarum* NK181 (T4), and *Lactobacillus delbrueckii* KU200171 (T5). The T5 sample showed high antioxidant activities (86.5 ± 0.3% and 39.3 ± 1.0% in DPPH and ABTS assays, respectively). The T4 sample had the highest ACE inhibitory activity (51.3 ± 10.3%). In the case of sensory evaluation, the T4 and T5 samples did not show a significant difference (*p* > 0.05) compared to the reference group. These results suggest that *L. plantarum* NK181 and *L. delbrueckii* KU200171 can be used in the food industry especially dairy to improve health benefits for hypertensive patients.

## 1. Introduction

The etymology of probiotics is derived from the Greek word meaning “pro-life” [1]. The definition of probiotics is “a preparation of or a product containing viable, defined microorganisms in sufficient numbers, which alter the microflora in a compartment of the host and by that exert beneficial health effects in this host.” [2]. 

The functional effects of probiotics have been extensively studied. Oelschlaeger [3] reported that the effects of probiotics might be classified into three modes of action. First, probiotic effects can be based on actions that affect microbial products such as toxins and host products such as bile salts and food ingredients. Second, probiotics can also directly affect other microbes, symbiotic and/or pathogenic microbes. Finally, Probiotics can modulate the host’s defenses, including the immune system. This mode of action is of paramount importance not only for the prevention and treatment of infectious diseases but also for the treatment of chronic inflammation of the digestive tract or parts thereof. In addition, probiotics have anticancer properties [4]. According to Lee et al. [5], *Lactobacillus plantarum* KU 200656 isolated from kimchi showed anti-pathogenic effects. In addition, *L. fermentum* strains may have potential cholesterol-lowering effects [6], and Liu et al. [7] reported that probiotics have a role in reducing risks associated with cardiovascular disease. 

Yogurt is one of the most general fermented foods worldwide [8]. It is a coagulated milk product from lactic acid bacteria (LAB) fermentation and contains proteins, such as casein [9]. Additionally, yogurt has good health benefits because of its bioactive protein hydolysates produced during fermentation [10,11]. In particular, yogurt has been shown to have an effect on diseases such as chronic illness, heart diseases, cancer, and diabetes [12,13]. However, these functional effects differ according to the variety of probiotic strains used.

Cardiovascular diseases, such as hypertension, often occur due to oxidative stress caused by various free radicals [14]. In particular, lung angiotensin-converting enzyme (ACE) plays an important role in blood pressure regulation, and inhibition of this enzyme promotes vasodilator production and sympathetic nervous system regulation. [15]. Recently, many pharmaceutical treatments have been developed to treat these chronic diseases. However, these methods often lead to treatment resistance. Therefore, there is a trend toward finding natural substances with relatively few side effects [16]. Fermented milk is considered a food that can have a positive effect on chronic diseases such as high blood pressure. In fact, many studies have shown that probiotics in fermented foods have an ACE inhibitory effect as well as antioxidant effects [10,17,18,19].

Thus, the purposes of this study were (1) to evaluate functional yogurt with ACE inhibition and antioxidant effects, and (2) to perform sensory evaluation of the products. In this study, we used the probiotic *Lacticaseibacillus rhamnosus* GG KCTC 12202BP, *Lactiplantibacillus plantarum* KU15003, *Lactiplantibacillus plantarum* KU15031, *Lactiplantibacillus plantarum* NK181, and *Lactobacillus delbrueckii* KU200171.

## 2. Materials and Methods

### 2.1. Microorganisms and Culture Conditions

The strains used in this study were *Lacticaseibacillus rhamnosus* GG KCTC 12202BP as a commercial reference strain, *Lactiplantibacillus plantarum* KU15003 and *Lactiplantibacillus plantarum* KU15031 (both isolated from kimchi), *Lactiplantibacillus plantarum* NK181 isolated from jeotgal (traditional Korean fermented food), and *Lactobacillus delbreuckii* KU200171 isolated from kimchi. These probiotics were isolated as described by Lee et al. [20]. All strains were cultured and maintained in de Man, Rogosa, and Sharpe (MRS) broth (BD BBL, Franklin Lakes, NJ, USA) at 37 °C.

### 2.2. Yogurt Sample Preparation

The yogurt was made with the method of Sah et al. [16] with few modifications. Briefly, six batches of milk bases were prepared by reconstitution with skim milk powder (12% *w*/*w*, Seoul Milk, Seoul, Korea), 2% (*w*/*w*) FOS (C_6_H_10_O_5_)_n_ (*n* > 10) (Cheil Jedang, Seoul, Korea) and water. The samples were pasteurized at 90 °C for 10 min. After cooling to 40 °C, all the mixtures were inoculated with a commercial LAB mixture (0.01% *w*/*w*, ABT-B; Culture Systems, Inc., Mishawaka, IN, USA). Except for the control (C), probiotics (colony count of the inoculum, 10^7^ CFU/mL) were inoculated to mixtures, respectively. Fermentation was terminated when the pH reached 4.4 ± 0.1. The yogurts were gently cooled and stored at 4 °C for a day. After the samples were ripened, all experiments were performed. The samples were named as: C, control with only commercial starter culture; T1, *L. rhamnosus* GG KCTC 12202BP; T2, *L. plantarum* KU15003; T3, *L. plantarum* KU15031; T4, *L. plantarum* NK181; and T5, *L. delbrueckii* KU200171.

### 2.3. pH and LAB Viability of Yogurt

The pH values of the samples were measured using a pH meter (InoLab pH 7110; Xylem Analytics Germany Sales. GmbH & Co. KG, Weilheim, Germany). Cell counts of all yogurt samples were determined during the fermentation period using the method described by Habibi et al. [15]. Each fermented milk was diluted with 0.1% peptone water every 3 h during fermentation. The diluted samples were inoculated at MRS agar at 37 °C. The MRS agar plates were cultivated for 48 h.

### 2.4. Preparation of Water-Soluble Extracts

Water-soluble extracts (WSEs) were prepared using the method described by Sah et al. and Kariyawasam and Lee [11,16] with a few modifications. The 30 g of yogurt was centrifuged at 14,000× *g* at 4 °C for 30 min to get supernatant. The supernatant filtered through a 0.45 μm membrane filter was freeze-dried and stored at less than −80 °C. In addition, the lyophilisates were dissolved with distilled water to analyze biological activity [11]. Through the Bradford assay using bovine serum albumin (0.1–1.2 mg/mL), the protein contents of dissolved WSE lyophilisates were fixed (0.5 mg of protein/mL).

### 2.5. Antioxidant Activity

The antioxidant activity of WSEs was confirmed using both 2,2-diphenyl-2-picrylhydrazyl (DPPH; Sigma–Aldrich Co., Steinheim, Germany) and 2,2-azinobis (3-ethylbenzothiazoline-6-sulfonic acid) di-ammonium salt (ABTS; Sigma–Aldrich Co., Steinheim, Germany) radical scavenging assays.

#### 2.5.1. DPPH Assay

The DPPH assay was performed as previously described [21] with slight modifications. WSE was mixed with 750 μL of 100 μM DPPH. The reaction was played in the dark at 25 °C for 20 min. The absorbance was measured at 517 nm using a spectrophotometer (Optizen 2120 UV; Mecasys Co., Ltd., Daejeon, Korea). Each sample was tested in triplicate. The DPPH radical scavenging activity was computed using the following formula:DPPH radical scavenging activity (%) = (1 − A_sample_/A_control_) × 100(1)

A_control_ and A_sample_ are the absorbances of the control (distilled water) and WSE, respectively.

#### 2.5.2. ABTS Assay

The ABTS assay was performed by the modified method of Yang et al. [22]. The solution was prepared with 14 mM ABTS and 5 mM K_2_S_2_O_8_; the solution was stored at 25 °C for 18 h with light blocked.

The prepared solution was diluted with 0.1 M sodium phosphate buffer (pH 7.4) until its absorbance approached 0.7 ± 0.05 at 734 nm. The 20 μL of WSE was added to 980 μL of ABTS solution. Then, the mixture was incubated in the dark at 25 °C for 20 min. Absorbances of all samples were measured in triplicate at 734 nm. The radical scavenge activity in ABTS assay was determined using the following formula:ABTS radical scavenging activity (%) = (1 − A_sample_/A_control_) × 100(2)

A_control_ and A_sample_ are the absorbances of the distilled water and WSE as samples, respectively.

### 2.6. Determination of ACE Inhibitory Activity

The ACE inhibitory activity of WSEs was confirmed by the methods of Cushman et al. and Kariyawasam and Lee [11,23] with slight modifications. Each 50 μL of WSE was added to 150 μL of 0.1 U/mL ACE solution and 100 μL of 0.1 M sodium borate buffer containing 0.3 M NaCl at pH 8.3. The mixture was kept at 37 °C for 10 min. ACE enzyme reagent used in this study was produced from rabbit lung and purchased from Sigma-Aldrich. A 50 μL of 5 mM hippuryl-L-histidyl-L-leucine (HHL; Sigma-Aldrich, St. Louis, MO, USA) in 0.1 M sodium borate buffer containing 0.3 M NaCl (pH 8.3) was added to the reaction mixture and stood at 37 °C for 30 min. The reaction was stopped by adding 250 μL of 1 N HCl. In addition, 200 μL of ethyl acetate was added to extract the liberated hippuric acid during the reaction. After mixing with a vortex mixer, the samples were centrifuged at 3000× *g* at 4 °C for 30 min. After the supernatants (ethyl acetate layer) were taken, ethyl acetate was completely evaporated using a water bath at 90 °C. The residue was dissolved in 1.0 mL of sodium borate buffer and refrigerated to 25 °C. All samples’ absorbances were measured in triplicate at 228 nm. The results were determined as inhibitory activity on ACE (%) using the following formula:Inhibitory activity on ACE (%) = (1 − A_sample_/A_control_) × 100(3)

A_control_ and A_sample_ are the absorbances of the control (buffer solution) and WSE, respectively.

### 2.7. Sensory Analysis

Following the results of antioxidant and ACE inhibitory activities (discussed in Section 3) only C, T1, T4, and T5 were used for sensory evaluation. Each yogurt was chilled at 4 °C for at least 24 h for ripening. Sensory analysis was performed by 31 trained panelists. Panels were trained with Sensory analysis—Methodology—Initiation and training of assessors in the detection and recognition of odors (ISO 5496:2006) [24]. The method was approved by the Institutional Review Board (approval number: IRB-SMU-C-2020-4-004, Korea). Quantitative descriptive analysis was performed to evaluate the differences in the sensory characteristics of the yogurt samples with probiotics. The color, taste, texture, flavor, and overall preference of each yogurt were measured. Water and plain bread were provided between the samples as a palette cleanser.

### 2.8. Statistical Analysis

Using IBM SPSS statistics 18 software, all data were explained with statistical analyses. Data were assessed using a one-way analysis of variance (ANOVA). Significant differences between the means were indicated by Duncan’s multiple range test at the *p* < 0.05 level. 

## 3. Results and Discussion

### 3.1. pH and Viability of LAB of Yogurt Samples

The pH values are shown in Figure 1. All the samples showed a decrease in pH during fermentation; the pH was reduced to 4.4 ± 0.1 in 9 h. Despite the difference in LAB composition, the pH values of yogurts with probiotics did not differ significantly (*p* > 0.05) compared to that of the yogurt with only the commercial starter culture. The viable cell counts in yogurt during fermentation are shown in Figure 2; they ranged from 6.3 to 9.1 log CFU/mL. In the viability of LAB, none of the data showed significant values (*p* > 0.05). The probiotics have to remain viable in food products above a threshold level to confirm probiotic health benefits. The minimum level of viable LAB should be at least 10^6^ CFU/g [25]. All the results showed reasonable values at the end of fermentation. Therefore, these probiotics were determined to be suitable as fermented strains of yogurt and to have potential functional effects.

### 3.2. Antioxidant Activities of Yogurts

The results of antioxidant activities are shown in Figure 3.

The DPPH assay is a simple and accurate method for confirming antioxidant effects [26]. The antioxidant activities recorded via DPPH assay for C and T1–T5 were 62.84 ± 5.17%, 71.33 ± 6.81%, 72.72 ± 5.82%, 75.00 ± 3.38%, 80.77 ± 1.30%, and 86.49 ± 0.28%, respectively. The IC_50_ values of C and T1–T5 were 0.38 mg/mL, 0.32 mg/mL, 0.32 mg/mL, 0.30 mg/mL, 0.29 mg/mL and 0.28 mg/mL, respectively.

The ABTS assay was also used to measure antioxidant activity. It measures the degree of inhibition of ABTS cations through the color change of a solution in the presence of antioxidants [27]. The antioxidant activities using the ABTS assay for C and T1–T5 were 37.65 ± 0.67%, 38.80 ± 0.62%, 38.00 ± 0.43, 38.25 ± 0.34%, 38.50 ± 0.41%, and 39.31 ± 0.99%, respectively. The IC_50_ values of C and T1–T5 were 0.66 mg/mL, 0.65 mg/mL, 0.65 mg/mL, 0.64 mg/mL, 0.65 mg/mL, and 0.64 mg/mL, respectively. The yogurt with *L. delbrueckii* KU200171 exhibited higher radical scavenging potency in all experiments than the yogurt with other probiotic strains. In this study, two methods, DPPH and ABTS, were used to evaluate antioxidant activity. The radical scavenging ability of antioxidants for DPPH and ABTS radicals may differ due to differences in solubility and diffusivity in the reaction medium. In addition, DPPH acts as an oxidative substrate and a reaction indicator. Spectral interference problems can easily arise. On the other side, ABTS is soluble in both aqueous and organic media. The radical scavenging activity of hydrophilic and lipophilic antioxidants can be evaluated using this method [28]. For this reason, it is considered that there is a difference between the results of DPPH and those of ABTS; in DPPH, T5 had a value about 10% higher than that of the C, and there was a significant difference in ABTS, but it had a similar value.

Milk proteins have been known as potential materials of biologically active peptides [29]. Donkor et al. [19] showed differences in bioactive peptides between yogurt starter cultures and probiotics.

Because of bioactive peptides released during fermentation, the antioxidant activities of WSEs were shown. Furthermore, the diverse antioxidant activity of yogurt WSE suggests that its radical scavenging activity depends on the difference of LAB strains and their individual enzyme patterns [11]. Thus, the antioxidant capacity of hydrolysates for the same substrate depends on the type of enzyme from the LAB, as specific proteases are complex in the hydrolysis of specific peptide bonds [16]. Specifically, Kudoh et al. [30] found the amino acid sequence of milk protein in fermented milk with *L. delbrueckii* ssp. *bulgaricus*. These amino acid sequences affect radical scavenging activity. Therefore, the results of this study conform to those reported in previous studies.

### 3.3. ACE Inhibition Activities of Yogurt

The ACE inhibition activity of the WSEs is shown in Figure 4. The activities of C and T1–T5 were 26.06 ± 8.61%, 35.60 ± 3.78%, 41.36 ± 4.30%, 37.95 ± 4.21%, 51.32 ± 10.30%, and 45.09 ± 4.58%, respectively. The IC_50_ values of C and T1–T5 were 2.28 mg/mL, 0.89 mg/mL, 0.68 mg/mL, 0.79 mg/mL, 0.48 mg/mL, and 0.64 mg/mL, respectively. The IC_50_ values were lower in the group to which probiotics were added compared to C. This suggests that yogurt with probiotics has an ACE inhibitory effect. Donkor et al. [19] reported that yogurts with probiotics had higher values than the control group. In this study, T4 and T5 showed significantly higher ACE inhibitory activities than C (*p* < 0.05; Figure 4). ACE converts angiotensin I to angiotensin II, which is involved in maintaining high blood pressure [31]. The bioactive peptides were known to have ACE inhibitory activities [18]. Cavalheiro et al. [32] reported that the higher the protein content in fermented milk, the higher the ACE inhibitory effect. Yogurt or cheese fermented with various LAB-released ACE inhibitory peptides with specific amino acid sequences [33]. Moreover, Vasiljevic and Snah [34] reported that β-casein hydrolysates by *L. delbrueckii* spp. *bulgaricus* had ACE inhibitory effects. As such, it can be seen that the hydrolyzed proteins made through probiotics exhibit an ACE inhibitory effect. Kariyawasam et al. [11] synbiotic yogurt had a higher ACE inhibitory activity than non-synbiotic yogurt. In this study, all samples were made with skim milk powder and fructooligosaccharide. Habibi et al. [15] suggested that the ACE inhibitory activities are influenced by fat content; that is, non-fat yogurt had higher ACE inhibitory activity than that containing fat. Habibi et al. [35] suggested that ACE inhibition effects depend on the presence of prebiotics. When yogurt contained prebiotics, the ACE inhibitory activities were higher than those of other samples [35]. Erkaya-Kotan [36] concluded that the degree of enzyme activity inhibition was correlated with the phenol content as well as the prebiotic content using orange fiber. Furthermore, soy yogurt with probiotics was shown to have a substantial increase in ACE inhibitory activity compared with the control samples produced by commercial yogurt culture [37]. As such, it can be seen that ACE inhibition is affected not only by probiotics but also by several other factors. Based on these previous studies, we attempted to obtain a high ACE inhibitory effect through the addition of prebiotics and fermentation using only skim milk powder. In this study, T4 with *L. plantarum* NK181 and T5 with *L. delbrueckii* KU200171 had a significantly higher (*p* < 0.05) ACE inhibitory activity compared to the C. These results are considered to be dependent on previous studies. Based on these results, *L. plantarum* NK181 and *L. delbrueckii* KU200171 were considered good probiotics for dairy products with ACE inhibitory activities.

### 3.4. Sensory Evaluation

Based on the higher antioxidant and ACE inhibitory activities of T4 and T5 compared to those of the other samples, they were selected as test samples. Because the C was prepared only with a starter and T1 was treated with *L. rhamnosus* GG, which is a commercial probiotic strain, these two groups were considered as reference groups. All samples were subjected to sensory evaluation for color, taste, texture, flavor, and overall acceptability. The results are presented in Table 1. For all sensory evaluations, except for color, T4 and T5 were given higher values than C. However, there were no significant differences between all samples (*p* > 0.05). Karimi et al. [25] reported the expression of various flavors and tastes in fermented foods according to various probiotics. For example, *L. rhamnosus* SP3 expressed a fruity flavor in cheese. Although there is a slight difference, compared to C, the reason that the yogurt with probiotics showed good results in sensory evaluation is thought to be due to the influence of the strains. It is considered that this functional yogurt is suitable for sensory evaluation. In addition, the stability of *L. plantarum* NK181 and *L. delbrueckii* KU200171, such as bile resistance, acid resistance, and intestinal adhesion ability, has been confirmed in previous studies [20,38]. Therefore, these probiotics are considered useful LAB in the dairy industry.

## 4. Conclusions

This study confirmed the functionality of yogurt fermented using various probiotics. Compared with C, all probiotic yogurts showed antioxidant activities, and ACE inhibitory effects, especially T4 treated with *L. plantarum* NK181 and T5 treated with *L. delbrueckii* KU200171 showed significantly higher activities (*p* < 0.05) than any other groups tested in this study. Sensory evaluation of those two samples showed no significant difference when compared with the commercial group. Recently, with increasing interest in promoting health through food, probiotics with inhibitory effects on angiotensin-converting enzymes are becoming attractive to the functional food industry for preventing adult chronic diseases, especially for hypertension patients. Furthermore, it is expected that this study will be useful in the application of functional fermented dairy materials in the pharmaceutical industry.

## Figures and Tables

**Figure 1 foods-10-02324-f001:**
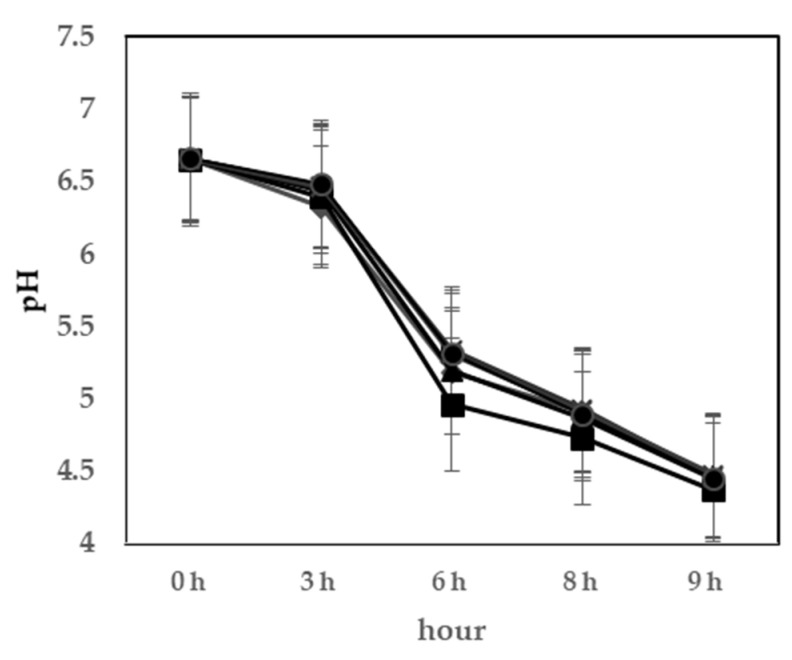
Changes in pH during fermentation at 40 °C. ◆, control yoghurt without probiotics (C); ■, *L. rhamnosus* GG KCTC 12202BP(T1); ▲, *L. plantarum* KU15003 (T2); ✕, *L. plantarum* KU15031(T3); ✳, *L. plantarum* NK181(T4); ●, *L. delbrueckii* KU200171(T5).

**Figure 2 foods-10-02324-f002:**
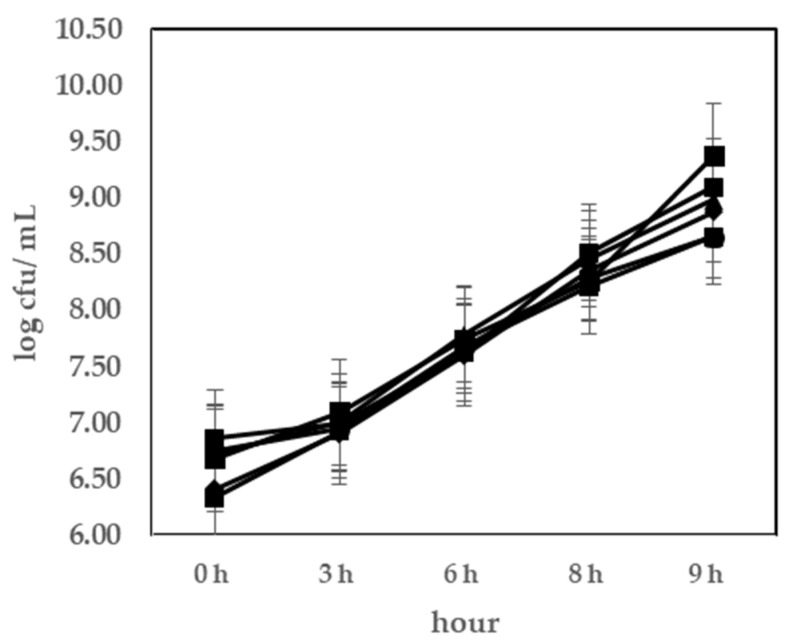
LAB bacterial cell numbers during fermentation at 40 °C. ◆, control yoghurt without probiotics (C); ■, *L. rhamnosus* GG KCTC 12202BP(T1); ▲, *L. plantarum* KU15003 (T2); ✕, *L. plantarum* KU15031(T3); ✳, *L. plantarum* NK181(T4); ●, *L. delbrueckii* KU200171(T5).

**Figure 3 foods-10-02324-f003:**
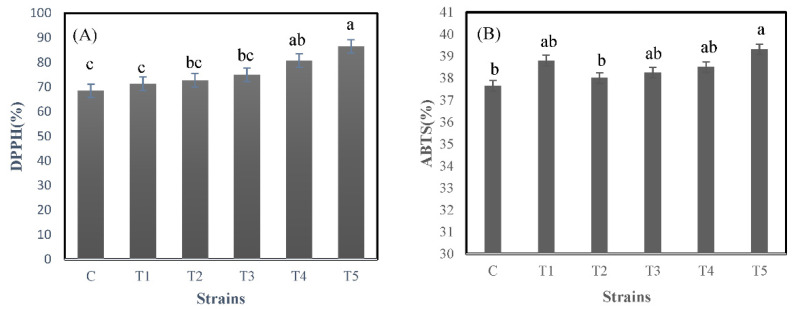
(**A**) DPPH and (**B**) ABTS radical scavenging assays of the water-soluble extracts (WSEs) from yogurt. C, control yogurt without probiotics; T1, *L. rhamnosus* GG KCTC 12202BP; T2, *L. plantarum* KU15003; T3, *L. plantarum* KU15031; T4, *L. plantarum* NK181; T5, *L. delbrueckii* KU200171. Means from different yogurt expressed by lowercase letters (a–c) are significantly different (*p* < 0.05).

**Figure 4 foods-10-02324-f004:**
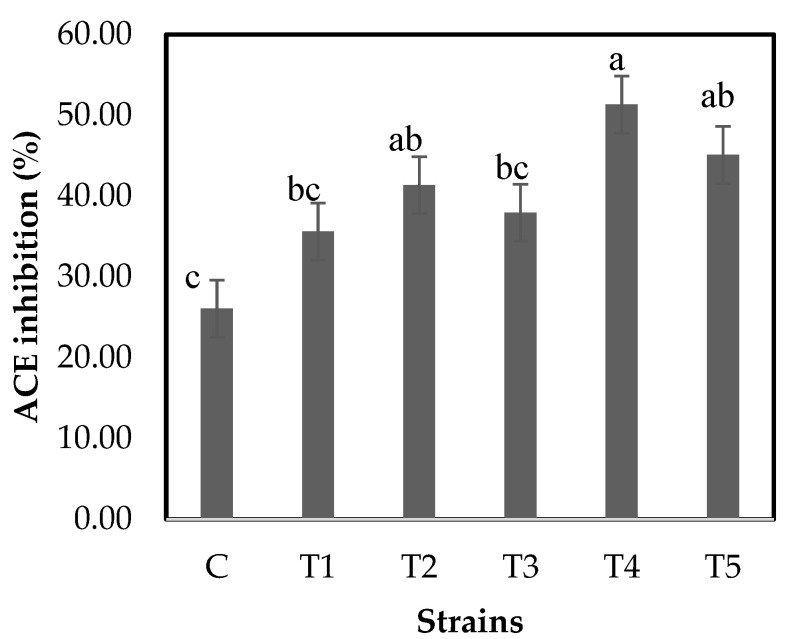
ACE inhibitory activity of the water-soluble extracts (WSEs) from yogurt. ABT-B, control yogurt without probiotics; T1, *L. rhamnosus* GG KCTC 12202BP; T2, *L. plantarum* KU15003; T3, *L. plantarum* KU15031; T4, *L. plantarum* NK181; T5, *L. delbrueckii* KU200171. Means from different yogurt expressed by lowercase letters (a–c) are significantly different (*p* < 0.05).

**Table 1 foods-10-02324-t001:** Consumer panelist ratings for the yogurt samples on a 7-point scale.

Sensory Evaluation	Samples			
	C	T1	T4	T5
Color	5.71 ± 1.35	5.61 ± 1.33	5.39 ± 1.31	5.48 ± 1.23
Taste	4.10 ± 1.33	4.19 ± 1.35	4.58 ± 1.31	4.55 ± 1.39
Texture	4.80 ± 1.40	5.19 ± 1.11	5.29 ± 1.30	5.06 ± 1.50
Flavor	4.90 ± 1.04	5.39 ± 0.99	5.42 ± 0.99	5.39 ± 0.88
Overall	4.16 ± 1.37	4.58 ± 1.20	4.61 ± 1.36	4.74 ± 1.29

Values are means ± standard deviation (*n* = 31); Means are not significantly different (*p* > 0.05). C, control yogurt without probiotics; T1, *L. rhamnosus* GG KCTC 12202BP; T4, *L. plantarum* NK181; T5, *L. delbrueckii* KU200171.

## Data Availability

Data sharing is not applicable.

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
