# Peer review of "Antioxidant and Angiotensin-Converting Enzyme (ACE) Inhibitory Activities of Yogurt Supplemented with Lactiplantibacillus plantarum NK181 and Lactobacillus delbrueckii KU200171 and Sensory Evaluation"

_foods, 2021, doi:10.3390/foods10102324_

Round 1

Reviewer 1 Report

General Comments.

The research on bioactive peptides is always desirable because of their importance. Bioactive peptides obtained for natural sources can be effective medicines and nutraceuticals The research described in the manuscript entitled: Antioxidant and Inhibitory Activities of Yogurt Supplemented with Lactiplantibacillus plantarum NK181 and Lactobacillusdelbrueckii KU200171 on Angiotensin-converting Enzymes and their Sensory Evaluation, was carried out correctly. However issues relating to the introduction and, above all, the description of results and discussion of results should be developed.

Specific Comments.

  1. The title should be rewritten on: Antioxidant and Angiotensin-Converting Enzyme (ACE) Inhibitory Activities of Yogurt Supplemented with Lactiplantibacillus plantarum NK181 and Lactobacillus delbrueckii KU200171 and Sensory Evaluation
  2. The Inhibitory activity was measured with the use ACE from rabbit lungh (Sigma) it contains only one enzyme, so please change plural Angiotensin-converting Enzymes (ACEs) on Angiotensin-converting Enzyme (ACE).
  3. Abstract: Please add „antioxidant activity” to sentence at line 12.
  4. Introduction: the positive health effects of probiotics in milk products come from many things, however two directions can be indicated: 1. benefits related to secretion by the probiotic cells of compounds such as: toxins for pathogenic microflora (anti-pathogenic effects) compounds stimulating production of Interferon gamma (immunostimulating effect) and enzymes (deconjugation of bile acids). 2. benefits related to impact of probiotics on milk compounds such as: fermentation of lactose (reducing intolerance connected with lactase deficiency) and proteolytic degradation of main milk proteins leading to obtain bioactive peptides (immunostimulating, antioxidant, ACE-inhibitory, anticancer effects). In my opinion text at lines 30-41 should be organized.
  5. The sentence lines 37-38: It should be „bioactive peptides” or „bioactive protein hydrolysates” instead of „bioactive peptide hydrates”
  6. The purpose 1 of study (lines 52-53): „… to select probiotics with healthful benefits (especially ACE inhibitory effect) and isolated from fermented foods such as kimchi or jeotgal…” suggests that the Authors somehow isolated and selected the best strains of probiotics for the production of yoghurt with antioxidant and ACE inhibitory activity. However, there is no isolation method and no indicators on the basis of which 6. probiotics have been selected. What was the selection criterion? Did the Authors measure the proteolytic activity of enzymes produced by selected strains? or did they produce trial yoghurts prior to actual production?
  7. Methodology Line 72 instead of „sterilized” should be” pasteurized”
  8. Please to explain why the antioxidant activity was determined on two free radicals? The mechanism of peptide activity is the same in both cases (ability to scavenge free radicals), wouldn't it be better to investigate besides free radical scavenging activity, another mechanism of action such as metal ion chelation, reduction of metal ion oxidation state, inhibition of oxidation propagation (on the example with the use linoleic acid)?
  9. Was the protein content in the obtained lyophilisates determined? if so, what the method was used?
  10. What were the lyophilisates dissolved in before determining the biological activity?
  11. In fact, the IC50 is used to use to determine the inhibitory activity. The IC50 value is estimating from a dose response curve of an inhibitor versus the ACE activity. However, the inhibitory activity measured for only one protein concentration may be random. Did the Authors think about determining IC50 at least for selected 1-2 trials?
  12. The short description of method for LAB viability of yoghurt is missing.
  13. The sentence at lines 189-190: „When DPPH solution is mixed with a substrate acting as a hydrogen atom donor, a stable 189 non-radical form of DPPH is obtained with an immediate change in the violet color to 190 yellow [22].”is not necessary.
  14. The Authors showed (lines 191-197) that the DPPH scavenging activity of specific samples is higher by min. 10% than for control. In the case of ABTS, the activity is at the same level. Please explain.
  15. The ACE inhibitory activity was measured for yogurts, The Authors did not purify and identify the peptides present in obtained yoghurts, therefore citing the peptide sequence is not appropriate (lines 239-241). Instead, a request to search for works on whole milk hydrolysates, including yoghurts, and to compare the levels of biological activity.
  16. Lines 237-251 commentary on the influence of other factors such as fat, inulin makes sense when the levels of activity from the cited works and those obtained by the authors are compared.
  17. In general, the coverage of the results and the discussion are too short.

Author Response

I sent a response letter as an attached file.

Reviewer 2 Report

Attached a file with comments

Author Response

(The authors gave the same response as above.)

Round 2

Reviewer 2 Report

Thanks for the review.

This manuscript is a resubmission of an earlier submission. The following is a list of the peer review reports and author responses from that submission.